# A comparison of the prevalence of and modifiable risk factors for cognitive impairment among community-dwelling Canadian seniors over two decades, 1991–2009

**Batholomew Chireh**[1]*, **Carl D'Arcy**[2]

**1** Saskatchewan Health Quality Council, Saskatoon, Saskatchewan, Canada, **2** Department of Psychiatry and School of Public Health, University of Saskatchewan, Saskatoon, Canada

* bac563@mail.usask.ca

## Abstract

### Background

The prevalence of cognitive impairment or dementia is of public health concern globally. Accurate estimates of this debilitating condition are needed for future public health policy planning. In this study, we estimate prevalence and modifiable risk factors for cognitive impairment by sex over approximately 16 years.

### Methods

Canadian Study of Health and Aging (CSHA) baseline data conducted between 1991–1992 were used to measure the prevalence of cognitive impairment and dementia among adults aged 65+ years. The standard Modified Mini-Mental State Examination (3MS) was used for the screening test for cognitive impairment. We compared the CSHA data with Canadian Community Health Survey–Healthy Aging (CCHS-HA) conducted between 2008–2009. The CCHS-HA used a four-dimension cognitive module to screen for cognitive impairment. Only survey community-dwelling respondents were included in the final sample. After applying exclusion criteria, final samples of (N = 8504) respondents in the CSHA sample and (N = 7764) respondents for CCHS–HA sample were analyzed. To account for changes in the age structure of the Canadian population, prevalence estimates were calculated using age-sex standardization to the 2001 population census of Canada. Logistic regression analyses were used to examine predictors of cognitive impairment. A sex stratified analysis was used to examine risk factors for cognitive impairment in the survey samples.

### Results

We found that prevalence of cognitive impairment among respondents in CSHA sample was 15.5% in 1991 while a prevalence of 10.8% was reported in the CCHS–HA sample in 2009, a 4.7% reduction [15.5% (CI = 14.8–16.3), CSHA *vs* 10.8% (CI = 10.1–11.5), CCHS–HA]. Men reported higher prevalence of cognitive impairment in CSHA study (16.0%) while women reported higher prevalence of cognitive impairment in CCHS–HA (11.6%). In the

**Data Availability Statement:** Data used for this analysis are not publicly available. The Canadian Study of Health and Aging (CSHA) study was a national, multicenter epidemiological study of

dementia among people aged 65 years or older data. The Canadian Community Health Survey—Healthy Aging - Cognition (CCHA-HA) component was also a national cross-sectional study conducted among adults aged 45+ years. Access to the CSHA dataset was provided by the Canadian Study of Health and Aging, available upon application at (http://www.csha.ca/). Data access to the CCHS-HA Cognition Master File was provided by Statistics Canada through their Research Data Centre's program that provides data access to Master File survey data to bona fide researchers upon application at (https://crdcn.org). The authors had no special access privileges to the data.

**Funding:** The University of Saskatchewan School of Public Health provided support for this study in the form of a scholarship awarded to BC.

**Competing interests:** The authors have declared that no competing interests exist.

multivariable analyses, risk factors such as age, poor self-rated health, stroke, Parkinson's disease, and hearing problems were common to both cohorts. Sex differences in risk factors were also noted.

## Conclusions

This study provides suggestive evidence of a potential reduction in the occurrence of cognitive impairment among community-dwelling Canadian seniors despite the aging of the Canadian population. The moderating roles of improved prevention and treatment of vascular morbidity and improvements in the levels of education of the Canadian population are possible explanations for this decrease in the cognitive impairment.

## Introduction

Dementia is more prevalent at older ages though it is not an inevitable consequence of aging [1]. In Canada in 2016, the estimated 5.9 million senior's population (65+ years) outnumbered the 5.8 million children (0–14 years) for the first time in the country's history [2]. Also, among the seniors, the percentage aged 80 or older continues to grow, as does the number of centenarians. These trends suggest that a rise in the prevalence of dementia can be anticipated [2].

Cognitive impairment is an intermediate state between normal aging and dementia. It signifies the transitional zone between normal cognitive function and clinically probable Alzheimer's disease (AD). People with cognitive impairment, have less severe cognitive deficits than those with dementia, and their normal daily function and independence while comprised are generally maintained. It is a chronic condition that is seen as a precursor to dementia in up to one-third of cases depending on age [3–5].

Research regarding global incidence and prevalence of cognitive impairment and dementia are mixed. While some studies are of the view that cognitive impairment or dementia is stable or on the decline, others report an increase in its incidence and prevalence. An earlier study reported in the Lancet projected dementia prevalence in North America to increase from 3.4 million in 2001 to 5.1 million whilst that of Europe to increase from 4.9 million in 2001 to 6.9 million in the year 2020 [6]. A Canadian report titled The Rising Tide by the Alzheimer Society, Canada [7] reported that dementia was on the increase in the general population. Also, a study conducted in the Canadian province of Alberta revealed increasing trends in dementia among the aboriginal population of the province [8]. However, another recent study in the neighboring province of Saskatchewan reported simultaneously a decreasing trend in incidence and an increasing trend in the prevalence of dementia [9]. The Canadian Chronic Disease Surveillance System using administrative data which includes community-dwelling and institutional populations shows a decreasing incidence of dementia between 2000 and 2016 (the latest year available) while the prevalence rate initial increase appears to have plateaued [10].

Also, in other Western countries, it has been reported that despite a declining trend in the age-specific incidence of dementia, the prevalence of cognitive impairment and dementia continue to grow along with the increase of life expectancy, as well as the associated burden in financial and social domains to the healthcare system [11–14]. However, three recent studies published in Lancet found that dementia prevalence had stabilized and was on the decline in Western Europe despite aging populations [15–17]. These authors believe that the reported reduction in dementia is a result of improved educational attainment, better prevention, and

treatment of vascular and chronic conditions. For cognitive impairment or dementia, other modifiable risk factors include tobacco smoking, alcohol consumption, exercise, fruit and vegetable consumption, anti-hypertensive medication, psychological and emotional health [18–20]. Non-modifiable risk factors such as age, sex, and apolipoprotein E-epsilon 4 allele (ApoE4) have also been reported [20, 21]. A recent Lancet Report [22] lists twelve risk factors for which there is scientific evidence of dementia occurring in various stages of the life cycle. They are less education, hypertension, hearing impairment, smoking, obesity, depression, physical inactivity, diabetes, low social contact, excessive alcohol consumption, traumatic brain injury, and air pollution. The report recommends that prevention action be taken on each of these risk factors.

The consequences of cognitive impairment are the increasing likelihood of dementia. A U. S. study estimated about 46% of people with cognitive impairment develop dementia within 3 years, compared to 3% of the age-matched population who were not cognitively impaired initially going on to develop dementia [23]. It has also been reported that seniors with cognitive impairment have a higher risk of avoidable injuries, hospitalization, and mortality [24–28]. A Canadian study reported that seniors (65+ years) are 1.5 times more likely than those aged 20–64 to present at an emergency department (ED) to seek health care including care for cognitive health issues [29]. Consistent with those findings, a U.S study also reported that cognitive impairment related incidents represent over 20% of emergency department consultations by seniors in the country [30].

Although point prevalence studies in cognitive impairment or dementia have been conducted in Canada, few have investigated the national prevalence over time and little is known as to whether modifiable risk factors of cognitive impairment have changed over the years in the context of an increasingly aging population. Better knowledge of the changing occurrence of cognitive impairment among different birth cohorts and changes over time in risk factors is required for informed policy planning.

The objectives of this study are to 1) estimate differences in the prevalence of cognitive impairment between two national population cohorts of community-dwelling seniors; 2) determine whether the risk factors of cognitive impairment changed over time; and 3) determine sex differences in common and unique risk factors for cognitive impairment in each study cohort and between cohorts.

## Materials and methods

### Data sources

The data sources examined here are two national surveys of older Canadians that are separated by a period of 16 years, namely, the Canadian Study of Health and Aging and the Canadian Community Health Survey Healthy Aging.

### Canadian Study of Health and Aging (1991–92, N = 10,263)

The Canadian Study of Health and Aging (CSHA) study was a national, multicenter epidemiological study of dementia among people aged 65 years or older data [31]. The first wave (CSHA-1) of this study started between 1991–1992 which drew a representative sample of people aged 65 years and older in Canada. Information was collected in person from an overall sample of 10,263 people aged 65 or over, evenly divided among the five geographic regions of Canada. Of the participants surveyed, 9,008 people were from the community and 1,255 lived in long-term-care institutions. Participants were assessed twice at 5-yearly intervals after the baseline. These in-person interviews broadly covered areas such as socio-demographics, health and well-being, disability, frailty, caregiving, dementia and cognitive impairment. Clinical

evaluation of the at risk and matching control subset of the larger sample also occurred. Data access was provided by the Canadian Study of Health and Aging [31]. For this analysis, the CSHA community sample of 9,008 participants was used.

## Canadian Community Health Survey–Healthy Aging (2008–09, N = 25,864)

We analyzed the second release of cross-sectional data from Statistics Canada's National Canadian Community Health Survey—Healthy Aging—Cognition component consisting of (N = 25,864) participants. The study targeted people aged 45 years and older using Computer-Assisted Personal Interviewing for data gathering. The topics covered included socio-demographics, well-being, and chronic diseases. The CCHS-HA did not sample full-time members of the Canadian Forces and residents of the three Canadian territories, Indian reserves, Crown lands, institutions, and some remote areas. The survey was weighted to represent the population of 45years of age and over living in the ten provinces of Canada between 2008-12-01 to 2009-11-30 [32]. The 2009 CCHS-HA cognition module analyzed here did not accept proxy responses resulting in a lower response rate of 62.4% compared to the 74.4% response rate recorded in the 2009 CCHS-HA main file. An initial subsample of participants aged 65 years and over, (N = 13, 306) was included in the current analysis.

Ethical approval was not required for this study because it was a secondary analysis of national health surveys already conducted by Statistics Canada. Written informed consent was obtained from all respondents, as well as ethics review approval for the original studies before they started. The ethics approval numbers of the original data could not be obtained because they were confidential national health surveys conducted by Statistics Canada and managed by the Social Sciences and Humanities Research Council (SSHRC). A written proposal was sent to Statistics Canada for access to the data which was subsequently vetted and approved by the Social Sciences and Humanities Research Council (SSHRC) with approval number (Application Id: 928488).

In both samples, the inclusion criteria were: (1) those who were 65 years and over; (2) participants who were screened or responded to questions concerning the outcome of interest, and (3) those without missing values on the variables of interest.

Fig 1 provides a detailed description of the criteria used to obtain the subsamples of the CSHA-Phase 1 and CCHS—HA- Cog-cohorts of respondents 65+ years. It also shows the screening process of the two samples using our inclusion and exclusion criteria. In the CSHA-1 sample, a total subsample of 8,504 was used in our analyses after excluding participants who did not meet our inclusion criteria. In the CCHS—HA Cog sample after all the missing values, not stated and those less than 65 years were excluded, a total figure of 7764 was used in the analyses (see Fig 1).

It should be noted that the CSHA cohort is composed of individuals born before 1926 and who no doubt experienced the Great Depression and WWII whereas the CCHS-HA 65 + cohort was born before 1943 had exposure to a substantially different social and economic environment.

## Measures

**Assessment of cognitive impairment in CSHA and CCHS-HA samples.**  In the (CSHA-Phase 1 1991) sample, the Modified Mini-Mental State Examination (3MS) was administered as a cognitive screen by a trained interviewer to identify respondents with cognitive impairments that merit a detailed clinical examination for specific dementias [33]. The 3MS is a widely used screening test for dementia and has a scoring system that ranges between (0–100) as the response scale for the participants. A cut-off point of ≤ 77 is conventionally used to

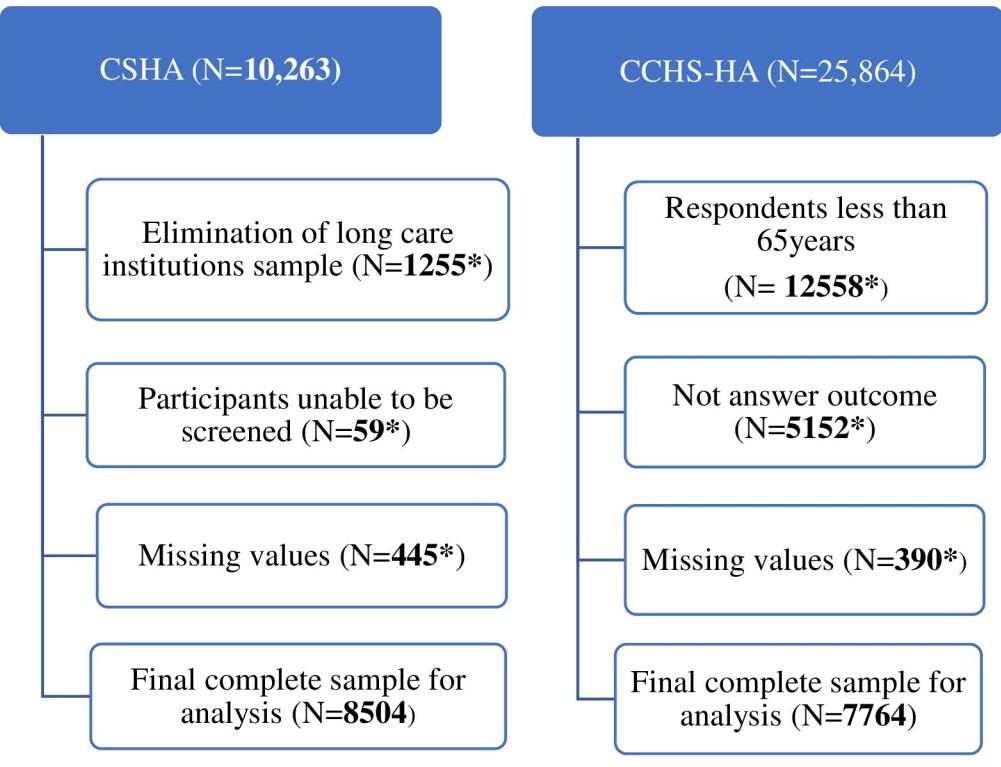

**Fig 1. Cohorts sample derivation.**

indicate potential cognitive impairment. Those scoring below 77 were considered cognitively impaired whilst those scoring above 77 were regarded as cognitively intact. In the CSHA sample, the outcome variable was derived by recoding all those participants with 3MS scores from 0 to 77 as ("yes" = 1) and those with 3MS scores from 78 to 100 as ('no" = 0).

In screening for dementia, the 3MS is a commonly used instrument in clinical settings. However, the CCHS—HA 2009 did not use the 3MS, but a cognition module also designed to screen for cognitive impairment. It measured four main domains of cognitive tasks: immediate and delayed recall which relate to memory function and the animal-naming and Mental Alternation Test which relate to executive function [34]. The outcome variable, cognitive impairment in the CCHS—HA 2009 cognition module is a derived variable based on the above domains of cognitive tasks. It sums up the number of component tasks where the respondent scored in the lowest cognitive functioning category. Therefore, the lower the functioning, the higher the score. The outcome variable has five categories ranging from (0–4), with a minimum value of 0 (indicating no impairment) and a maximum value of 4. For this analysis, Category 0 was recoded as ("no" = 0) to represent respondents without cognitive impairment, while categories (1–4) were collapsed and recoded as ("yes" = 1) to represent the presence of cognitive impairment, a conservative assumption and all other values were treated as missing.

**Predictors/Covariates.** Several modifiable and non-modifiable risk factors or covariates were assessed in both cohort samples. Both surveys collected information on the following predictors or covariates which were therefore included in the analysis: *gender* ("male" and

"female"), *age* (65–74, 75–84, 85+ years), *area of residence* ("rural" and "urban"), *educational level* ("Less than secondary", "Secondary graduation", "Some post-secondary graduation", "Post-secondary graduation"), *region of residence* ("Atlantic", "Quebec", "Ontario", "Prairies", "British Columbia"), *marital status* ("Married/common law", "Widowed/divorced/separated", "Single/never married"), *cultural or racial background* ("white" and "all other race/non-white"), *self-rated health* ("excellent/very good/good" and "fair/poor"), *high blood pressure* ("yes" and "no"), *heart disease* ("yes" and "no"), *arthritis* ("yes" and "no"), *Parkinson disease* ("yes" and "no"), *diabetes* ("yes" and "no"), stroke ("yes" and "no"), *hearing problems* ("yes" and "no") and *vision problems* ("yes" and "no").

## Statistical analysis

We first examined the prevalence of cognitive impairment between the two cohorts using age-sex standardization comparison methods. We standardized the prevalence estimates using the Canadian 2001 population census data on age and sex. These estimates were used to generate cognitively impaired proportions and 95% confidence intervals (CI) in both samples. Age was categorized as 65–74 years, 75–84 years, and 85+ years.

The second phase of the analyses focused on the model building process to ascertain whether any sex differences in risk factors for cognitive impairment had changed over time. We performed multiple imputations to adjust for missing risk factor values and to prevent selection bias and loss of information. The imputations were generated using the chained equations procedure in STATA. All covariates and the outcome variable were included in the imputation model to ensure accurate estimation of missing values. After the imputation process, we retrieved all the missing values in both samples for the model building.

Logistic regression models were employed at the univariate analysis stage between each predictor and outcome in the two cohorts and reported unadjusted odds ratios (UOR), 95% confidence intervals (CI) and p-values. Predictors with unadjusted $p < 0.20$ were maintained for further use in the multivariable analysis [35].

In the multivariable model building, logistic regression models were also used to examine the association between predictor variables and the outcome, and a manual backward elimination process was used to remove insignificant variables one at a time. All other variables recording a significance level of $p < 0.05$ were retained in subsequent analyses. Significant potential confounders were also tested. Insignificant variables at the univariate analysis stage were also tested for confounding, they were not confounders and therefore were left out of the final model. Three logistic regression models were built: 1) multivariable analysis between predictor variables and outcome in the two samples 2) association between predictor variables and outcome in males 3) association between predictor variables and outcome in females. We also checked the overall significance of all the logistic regression models in both samples by using a likelihood ratio test. All statistical analyses were completed using Stata 14.

## Results

### Characteristics of the study population

The study included final unweighted subsamples of 8504 and 7764 of seniors (65+ years) for the survey years 1991/92 and 2008/09 respectively. Table 1 below shows the demographic characteristics of respondents for both study samples. In both samples, most of the respondents were females (59.7%) and (60.4%) respectively. The later CCHS-HA sample differs from the earlier CSHA sample in having a greater number of individuals from the Atlantic region of Canada, 25.9%, versus 20.1%; more individuals from the Prairies, 23.9% compared to 19.2%; but, a lower proportion from the British Columbia on the Pacific coast of Canada. The

**Table 1. Socio-demographics of CSHA (1991) and CCHS-HA (2009) samples.**

| Characteristics | CSHA | CCHS-HA |
|---|---|---|
| N | 8504 (100%) | 7764 (100%) |
| **Province** | | |
| Atlantic | 1709 (20.1%) | 2009 (25.9%) |
| Quebec | 1718 (20.2%) | 1460 (18.8%) |
| Ontario | 1709 (20.1%) | 1509 (19.4%) |
| Prairies | 1631 (19.2%) | 1852 (23.9%) |
| British Columbia | 1737 (20.4%) | 934 (12.0%) |
| **Gender** | | |
| Male | 3430 (40.3%) | 3078 (39.6%) |
| Female | 5074 (59.7%) | 4686 (60.4%) |
| **Age categories (years)** | | |
| 65–74 | 3759 (44.2%) | 4019 (51.8%) |
| 75–84 | 3535 (41.6%) | 2525 (32.5%) |
| 85+ | 1210 (14.2%) | 1220 (15.7%) |
| **Marital Status** | | |
| Married/common-law | 4363 (51.3%) | 3858 (49.7%) |
| Widowed/divorced/separated | 3562 (41.9%) | 3502 (45.1%) |
| Single/never married | 579 (6.8%) | 404 (5.2%) |
| **Area of residence** | | |
| Rural | 1196 (14.1%) | 1713 (22.1%) |
| Urban | 7308 (85.9%) | 6051 (77.9%) |
| **Education** | | |
| Less than secondary | 2444 (28.7%) | 2871 (37.0%) |
| Secondary graduation | 3619 (42.6%) | 1127 (14.5%) |
| Other post-secondary | 1292 (15.2%) | 420 (5.4%) |
| Postsecondary graduation | 1149 (13.5%) | 3346 (43.1%) |

CCHS-HA sample also had a greater number of respondents in the 65–74 age group compared to the CHSA sample, 51.8% versus 44.2%. It also had a larger number of respondents from rural areas than the CSHA cohort, 23.1% versus 14.1%. In terms of education, the more recent CCHS-HA had more respondents with *less than secondary education* compared to the older CSHA cohort, 37.0% vs 28.7%. But the CCHS-HA had substantially more individuals who have completed post-secondary graduation, 43.1% vs 13.5%, than the CSHA. Many participants in both samples were married or in common-law relationships (51.3%) and (49.7%) respectively. The majority of the respondents in both samples lived in urban areas (85.9% *vs* 77.9%).

## Age–sex standardized prevalence of cognitive impairment between 1991 to 2009

The standardized prevalence (adjusted for age and sex) of cognitive impairment as measured in this study showed a decrease from across time from 15.5% in the CSHA (1991–92) study to 10.8% in the CCHS–HA study (2008–09). An overall decrease of 4.7% during the 16 years.

Whereas men had a higher prevalence of cognitive impairment in CSHA study, women had a higher prevalence of cognitive impairment in CCHS–HA (with a prevalence of 16.0% for men in 1991–92 *vs* 15.1% for women in 2008–09). Between 1991 to 2009 men had a larger

**Table 2. Age–sex standardized prevalence of cognitive impairment for men and women aged 65+ years in 1991 and 2009, and differences between 1991 and 2009.**

| Characteristics | CSHA 1991 | CCHS–HA 2009 | Difference |
|---|---|---|---|
| | %(95%CI) | %(95%CI) | |
| **Men** | | | |
| 65–74 years | 10.7% (9.2–12.1) | 9.2% (7.8–10.5) | −1.5% |
| 75–84 years | 21.9% (19.7–24.1) | 8.4% (6.6–10.3) | −13.5% |
| 85+ years | 46.0% (40.9–51.2) | 6.3% (3.8–8.7) | −39.7% |
| **Women** | | | |
| 65–74 years | 12.6% (11.2–14) | 7.9% (6.7–9.1) | −4.7% |
| 75–84 years | 17.3% (15.7–18.9) | 11.8% (10.2–13.3) | −5.5% |
| 85+ years | 37.5% (34.2–40.8) | 7.5% (5.7–9.3) | −30% |
| **By sex, standardized to 2001 Canadian population census** | | | |
| Men | 16.0% (14.9–17.2) | 9.6% (8.6–10.7) | −6.4% |
| Women | 15.1% (14.2–16.1) | 11.6% (10.7–12.6) | −3.5% |
| Overall prevalence | 15.5% (14.8–16.3) | 10.8% (10.1–11.5) | −4.7% |

reduction in the prevalence of cognitive impairment compared to women, with a reduction of 6.4 percentage points for men *vs* 3.5 percentage point reduction for women.

Our analyses show a significant decrease in the prevalence of cognitive impairment in all age categories with the deceased being most pronounced among those 85+ with prevalence decreasing from 46. % and 37. % in men and women respectively in 1991–92 to 6.3% and 7.5% respectively in 2008–09. Cognitive impairment prevalence in the young-old (65–74 years) age group decreased marginally from 10.7% (95%CI = 9.2–12.1) in 1991–92 to 9.2% (95% CI = 7.8–10.5) for men in 2008–09. In the same vein, women recorded a slight decrease from 12.6% (95%CI = 11.2–14) in 1991–92 to 7.9% (95%CI = 6.7–9.1) in 2008–09. This translates into a percentage point decrease of 4.7 for women *vs* 1.5 for men. The age gradient for cognitive impairment prevalence was much less steep in the CCHS-HA cohort in contrast to the earlier CSHA cohort (Table 2).

## Characteristics associated with cognitive impairment between 1991 and 2009 (multivariable analysis)

In our multivariable logistic regression analysis, we found five shared risk factors for cognitive impairment between the two cohorts: *age, self-rated health, stroke, Parkinson's disease, and hearing problems* (Table 3). Not surprisingly, we found that the odds of developing cognitive impairment increases with increasing age. Compared to the young-old age groups (65–74 years) respondents in the oldest–old age groups (85+ years) were more likely to report cognitive impairment (OR = 6.63, p<0.001, CSHA *vs* OR = 2.19, p<0.001, CCHS–HA). Similarly, seniors who rated their health as poor in the CSHA cohort were 1.69 times (p<0.001) more likely to report cognitive impairment compared to 1.33 times (p<0.001) in the CCHS–HA cohort. Respondents suffering from stroke were more likely to report cognitive impairment compared to others (OR = 2.09, p<0.001, CSHA *vs* OR = 1.29, p<0.001, CCHS–HA). Also, the odds of reporting cognitive impairment were higher in respondents with Parkinson's disease compared to those without the disease (OR = 1.99, p = 0.002, CSHA *vs* OR = 1.34, p = 0.152, CCHS–HA). Also, people with hearing problems had 58% higher odds (p<0.001) of reporting cognitive impairment in the CSHA cohort compared to 54% higher odds (p<0.001) in the CCHS–HA cohort.

Our study found six shared *protective* factors for cognitive impairment: being female, being 'white', urban residence, a high level of education, having blood pressure, and having heart

**Table 3. Multivariable analysis of risk factors for cognitive impairment between CSHA 1991 and CCHS-HA 2009.**

| | CSHA 1991 | | CCHS-HA 2009 | |
| --- | --- | --- | --- | --- |
| Characteristics | OR, 95% CI | p-Value | OR, 95% CI | p-Value |
| **Gender** | | | | |
| Male | 1 | | 1 | |
| Female | 0.87 (0.75–0.99) | 0.049 | 0.90 (0.83–0.97) | 0.004 |
| **Age categories, years** | | | | |
| 65–74 | 1 | | 1 | |
| 75–84 | 2.37 (2.04–2.75) | <0.001 | 1.39 (1.28–1.51) | <0.001 |
| 85 and above | 6.63 (5.53–7.96) | <0.001 | 2.19 (1.99–2.41) | <0.001 |
| **Marital status** | | | | |
| Married/common law | 1 | | 1 | |
| Widowed/div./separated | 1.15 (0.99–1.33) | 0.065 | N/A | N/A |
| Single/never married | 1.55 (1.22–1.97) | <0.001 | N/A | N/A |
| **Race/ethnicity** | | | | |
| Non-white | 1 | | 1 | |
| White | 0.34 (0.22–0.53) | <0.001 | 0.54 (0.45–0.65) | <0.001 |
| **Educational level** | | | | |
| Less than secondary | 1 | | 1 | |
| Secondary graduation | 0.28 (0.24–0.32) | <0.001 | 0.77 (0.69–0.86) | <0.001 |
| Some post-secondary | 0.17 (0.14–0.21) | <0.001 | 0.65 (0.55–0.78) | <0.001 |
| Postsecondary graduation | 0.10 (0.08–0.14) | <0.001 | 0.88 (0.81–0.95) | 0.001 |
| **Area of residence** | | | | |
| Rural | 1 | | 1 | |
| Urban | 0.82 (0.70–0.96) | 0.014 | 0.90 (0.83–0.98) | 0.011 |
| **Self-rated Health** | | | | |
| Good health | 1 | | 1 | |
| Poor health | 1.69 (1.46–1.97) | <0.001 | 1.33 (1.22–1.45) | <0.001 |
| **High blood pressure** | | | | |
| No | 1 | | 1 | |
| Yes | 0.82 (0.71–0.93) | 0.003 | 0.93 (0.87–1.00) | 0.056 |
| **Heart disease** | | | | |
| No | 1 | | 1 | |
| Yes | 0.80 (0.70–0.92) | 0.002 | 0.94 (0.86–1.02) | 0.157 |
| **Stroke** | | | | |
| No | 1 | | 1 | |
| Yes | 2.09 (1.63–2.68) | <0.001 | 1.29 (1.09–1.53) | <0.001 |
| **Arthritis** | | | | |
| No | 1 | | 1 | |
| Yes | 0.74 (0.65–0.84) | <0.001 | N/A | N/A |
| **Parkinson** | | | | |
| No | 1 | | 1 | |
| Yes | 1.99 (1.29–3.06) | 0.002 | 1.34 (0.90–2.0) | 0.152 |
| **Diabetes** | | | | |
| No | 1 | | 1 | |
| Yes | N/A | N/A | 1.04 (0.96–1.15) | 0.407 |
| **Hearing problems** | | | | |
| No | 1 | | 1 | |
| Yes | 1.58 (1.36–1. 84) | <0.001 | 1.54 (1.40–1. 69) | <0.001 |

(*Continued*)

**Table 3.** (Continued)

| Characteristics | CSHA 1991 | | CCHS-HA 2009 | |
|---|---|---|---|---|
| | OR, 95% CI | p-Value | OR, 95% CI | p-Value |
| Vision problems | | | | |
| No | 1 | | 1 | |
| Yes | 1.35 (1.14–1. 60) | <0.001 | N/A | N/A |

disease. Females in 1991 were 13% (p = 0.049) less likely to report cognitive impairment compared to 10%(p<0.001) of females in 2009. Secondly, 'white' respondents were less likely to report impairment compared to their non–white counterparts (OR = 0.34, p<0.001, CSHA *vs* OR = 0.54, p<0.001, CCHS–HA). Participants who lived in an urban area were less likely to report cognitive impairment compared to their rural counterparts (OR = 0.82, p = 0.014, CSHA *vs* OR = 0.90, p = 0.011, CCHS–HA). Educational attainment was a protective factor for cognitive impairment in both cohorts. Compared to the less than secondary graduation, those who attained postsecondary education were much less likely to report cognitive impairment (OR = 0.10, p<0.001, CSHA *vs* OR = 0.88, p<0.001, CCHS–HA). We found a negative relationship between high blood pressure and cognitive impairment. Hypertensive respondents in the CSHA cohort had a 0.82 (p = 0.003) lower odds of reporting cognitive impairment compared to a 0.93 (p = 0.056) lower odds in the CCHS–HA cohort. Similarly, respondents with heart disease were less likely to report cognitive impairment (OR = 0.80, p = 0.002, CSHA *vs* OR = 0.94, p = 0.157, CCHS–HA). It is unclear whether those reporting high blood pressure and heart disease were being physically active and effectively treated for those conditions. Given the nature of the question (have you been told by a doctor that you have high blood pressure and/or heart disease?) and the fact that Canada has a universal publicly funded health care system, we assume that this is the case.

Additionally, we found contrasting risk factors for cognitive impairment between the two cohorts analyzed. These include *marital status, arthritis, diabetes, and vision problems*. Compared to married respondents, those who were single or never married were 1.55 times (p<0.001) more likely to report cognitive impairment in the CSHA cohort but this was not a risk factor in the CCHS–HA cohort. Also, diabetic respondents in the CCHS–HA cohort were 1.04 times (p = 0.407) more likely to report cognitive impairment but this was not a risk factor in the CSHA cohort. Besides, respondents with vision health problems were more likely to report cognitive impairment (OR = 1.35, <0.001) in the CSHA cohort but not in the CCHS–HA cohort. In contrast, arthritis was a protective factor in the CSHA cohort but was not a significant variable for the CCHS-HA cohort.

## Characteristics associated with cognitive impairment in the 1991–92 sample by sex

Six factors, age, marital status, self-rated health, stroke, hearing problems, and vision problems were risk factors for cognitive impairment in both sexes in 1991–92 (Table 4). Compared to the young-old age groups (65–74years) males in the oldest–old age group (85+ years) were 6.51 times (p<0.001) more likely to report cognitive impairment compared to 6.95 times (p<0.001) for females in the same age group. Relative to married respondents, single or never married males were 2.10 times (p<0.001) more likely to report cognitive impairment compared to 1.37 times (p = 0.042) for females. Both males and females with poor self-rated health were more likely to report cognitive impairment compared to those with good self-rated health (OR = 1.64, p<0.001, male *vs* OR = 1.67, p<0.001, female). Besides, participants suffering

**Table 4. Multivariable analysis for risk factors of cognitive impairment for CSHA 1991 cohort by sex.**

| | CSHA 1991 Sample | | | |
|---|---|---|---|---|
| | Males | | Females | |
| Characteristics | OR, 95% CI | p-Value | OR, 95% CI | p-Value |
| **Age categories, years** | | | | |
| 65–74 | 1 | | 1 | |
| 75–84 | 2.48 (2.00–3.08) | <0.001 | 2.32 (1.89–2.86) | <0.001 |
| 85 and above | 6.51 (4.87–8.70) | <0.001 | 6.95 (5.46–8.85) | <0.001 |
| **Marital status** | | | | |
| Married/common law | 1 | | 1 | |
| Widowed/div./separated | 1.10 (0.88–1.38) | 0.418 | 1.14 (0.94–1.39) | 0.193 |
| Single/never married | 2.10 (1.38–3.19) | <0.001 | 1.37 (1.01–1.87) | 0.042 |
| **Race/ethnicity** | | | | |
| Non-white | 1 | | 1 | |
| White | 0.41 (0.21–0.79) | 0.008 | 0.29 (0.16–0.53) | <0.001 |
| **Educational level** | | | | |
| Less than secondary | 1 | | 1 | |
| Secondary graduation | 0.33 (0.27–0.41) | <0.001 | 0.24 (0.20–0.29) | <0.001 |
| Some post-secondary | 0.22 (0.15–0.31) | <0.001 | 0.15 (0.11–0.19) | <0.001 |
| Postsecondary graduation | 0.09 (0.06–0.14) | <0.001 | 0.11 (0.08–0.16) | <0.001 |
| **Self-rated Health** | | | | |
| Good health | 1 | | 1 | |
| Poor health | 1.64 (1.31–2.06) | <0.001 | 1.67 (1.37–2.03) | <0.001 |
| **High blood pressure** | | | | |
| No | N/A | | 1 | |
| Yes | N/A | | 0.80 (0.67–0.95) | 0.010 |
| **Heart disease** | | | | |
| No | N/A | | 1 | |
| Yes | N/A | | 0.77 (0.64–0.92) | 0.004 |
| **Stroke** | | | | |
| No | 1 | | 1 | |
| Yes | 2.19 (1.54–3.11) | <0.001 | 1.89 (1.34–2.67) | <0.001 |
| **Arthritis** | | | | |
| No | 1 | | 1 | |
| Yes | 0.77 (0.64–0.93) | <0.001 | 0.73 (0.61–0.86) | <0.001 |
| **Parkinson** | | | | |
| No | 1 | | 1 | |
| Yes | 2.91 (1.58–5.37) | 0.001 | N/A | |
| **Hearing problems** | | | | |
| No | 1 | | 1 | |
| Yes | 1.71 (1.37–2. 12) | <0.001 | 1.51 (1.22–1. 86) | <0.001 |
| **Vision problems** | | | | |
| No | 1 | | 1 | |
| Yes | 1.33 (1.01–1. 75) | 0.045 | 1.40 (1.13–1. 72) | 0.002 |

from stroke were more likely to report cognitive impairment in both sexes. Compared to stroke-free participants, males were twice as likely (OR = 2.19, p<0.001) to report cognitive impairment than females (OR = 1.89, p<0.001). Also, males with hearing problems were 1.71 times (p<0.001) more likely to report depression compared to 1.51 times (p<0.001) for

females. Vision problems were also significantly associated with cognitive impairment in both sexes (OR = 1.33, p = 0.045, male *vs* OR = 1.40, p = 0.002, female).

We found three common protective factors of cognitive impairment between males and females. Cultural or racial background, educational level, and arthritis were the protective factors for cognitive impairment. Compared to "non-white", white males had a 59% (p = 0.008) lower odds of reporting cognitive impairment as against a 71% (p<0.001) lower odds in white females. Respondents with the highest educational level (postsecondary graduation) in both sexes were less likely to report cognitive impairment (OR = 0.09, p<0.001, male *vs* OR = 0.11, p<0.001, female) compared to the reference group. Arthritis was also a significant protective factor for males and females in the 1991 cohorts (OR = 0.77, p<0.001, male *vs* OR = 0.73, p<0.001, female).

Our study found high blood pressure and heart disease as the two unique protective factors for females only. Females with high blood pressure had a 20% (p = 0.010) lower odds of reporting cognitive impairment but not males. Similarly, females with heart disease had a 23% (p = 0.004) lower odds of reporting cognitive impairment but not for males. In contrast, we found Parkinson's disease a unique risk factor for males but not for females. Males with Parkinson's disease were 2.91 times (p<0.001) more likely to report cognitive impairment compared to females (Table 4).

## Characteristics associated with cognitive impairment in the 2008–09 sample by sex

We found four shared risk factors, age, self-rated health, stroke and hearing problems were positively associated with cognitive impairment in both males and females (Table 5). In comparison to the young-old age group (65–74 years) male seniors in the oldest-old age group (85 + years) were 1.99 times (p<0.001) more likely to report cognitive impairment compared to 2.27 times (p<0.001) for female seniors in the same age group. Male respondents with poor self-rated health had a 28% (p = 0.501) higher odds of reporting cognitive impairment compared a 36% (p<0.001) higher odds in females. Also, both males and females with stroke compared to those without stroke were more likely to report cognitive impairment (OR = 1.52, p<0.001, male *vs* OR = 1.10, p = 0.406, female). In addition, females with hearing problems were more likely to report cognitive impairment than males (OR = 1.50, p<0.001, male *vs* OR = 1.58, p<0.001, female).

Among males only, we found marital status and Parkinson's disease as unique risk factors for cognitive impairment. Compared to those who were married or in a common-law relationship, male respondents who were single or never married were more likely to report cognitive impairment (OR = 1.15, p = 0.282). Also, male respondents with Parkinson's disease were 1.84 times (p = 0.044) to report cognitive compared to those without the disease.

Among females only, we found diabetes and arthritis as unique risk factors for cognitive impairment. Compared to non-diabetic female respondents, females with diabetes were 1.10 times (p = 0.153) more likely to report cognitive impairment. Similarly, arthritis was also positively associated with cognitive impairment for females only (OR = 1.04, p = 0.414).

In summary, our findings concerning risk and protective factors among men and women and in both the CSHA and CCHS-HA study cohorts are summarized in Table 6.

## Discussion

The first objective of this study was to estimate the age-sex standardized prevalence of cognitive impairment between two-time points among community-dwelling Canadian seniors using population-based national surveys. We observed a reduction in cognitive impairment in

**Table 5. Multivariable analysis of risk factors for cognitive impairment for CCHS-HA 2009 cohort by sex.**

| | CCHS-HA 2009 | | | |
|---|---|---|---|---|
| | Males | | Females | |
| Characteristics | OR, 95% CI | p-Value | OR, 95% CI | p-Value |
| **Age categories, years** | | | | |
| 65–74 | 1 | | 1 | |
| 75–84 | 1.26 (1.11–1.43) | <0.001 | 1.48 (1.33–1.65) | <0.001 |
| 85 and above | 1.99 (1.69–2.34) | <0.001 | 2.27 (2.01–2.57) | <0.001 |
| **Marital status** | | | | |
| Married/common law | 1 | | 1 | |
| Widowed/div./separated | 1.23 (1.08–1.40) | 0.002 | N/A | N/A |
| Single/never married | 1.15 (0.89–1.48) | 0.282 | N/A | N/A |
| **Race/ethnicity** | | | | |
| Non-white | 1 | | 1 | |
| White | 0.57 (0.43–0.74) | <0.001 | 0.51 (0.39–0.67) | <0.001 |
| **Educational level** | | | | |
| Less than secondary | 1 | | 1 | |
| Secondary graduation | 0.78 (0.27–0.41) | 0.010 | 0.77 (0.67–0.88) | <0.001 |
| Some post-secondary | 0.75 (0.15–0.31) | 0.035 | 0.58 (0.46–0.73) | <0.001 |
| Postsecondary graduation | 0.91 (0.06–0.14) | 0.128 | 0.86 (0.77–0.95) | 0.005 |
| **Self-rated Health** | | | | |
| Good health | 1 | | 1 | |
| Poor health | 1.28 (1.12–1.47) | <0.001 | 1.36 (1.21–1.53) | <0.001 |
| **High blood pressure** | | | | |
| No | 1 | | 1 | |
| Yes | 1.04 (0.93–1.16) | 0.501 | 0.86 (0.78–0.95) | 0.002 |
| **Heart disease** | | | | |
| No | 1 | | 1 | |
| Yes | 0.99 (0.87–1.12) | 0.896 | 0.91 (0.81–1.02) | 0.101 |
| **Stroke** | | | | |
| No | 1 | | 1 | |
| Yes | 1.52 (1.18–1.96) | 0.001 | 1.10 (0.87–1.40) | 0.406 |
| **Arthritis** | | | | |
| No | 1 | | 1 | |
| Yes | N/A | N/A | 1.04 (0.95–1.14) | 0.414 |
| **Parkinson** | | | | |
| No | 1 | | 1 | |
| Yes | 1.84 (1.02–3.33) | 0.044 | N/A | N/A |
| **Diabetes** | | | | |
| No | 1 | | 1 | |
| Yes | N/A | N/A | 1.10 (0.97–1.25) | 0.153 |
| **Hearing problems** | | | | |
| No | 1 | | 1 | |
| Yes | 1.50 (1.30–1. 73) | <0.001 | 1.58 (1.38–1. 80) | <0.001 |
| **Vision problems** | | | | |
| No | 1 | | 1 | |

(*Continued*)

**Table 5.** (Continued)

| Characteristics | CCHS-HA 2009 | | | |
| --- | --- | --- | --- | --- |
| | Males | | Females | |
| | OR, 95% CI | p-Value | OR, 95% CI | p-Value |
| Yes | 0.85 (0.74–0. 97) | 0.015 | N/A | N/A |

We also found three shared protective factors of cognitive impairment among males and females. These include heart disease, education, and cultural or racial background. Male seniors with heart disease had slightly lower odds of 0.99 (p<0.896) of reporting cognitive impairment compared to 0.91(p = 0.101) lower odds in female seniors with heart disease. Relative to other groups, white males had a 43% (p<0.001) lower odds of reporting cognitive impaired compared to a 49%(p<0.001) lower odds among females.

Canada over 16 years (between 1991–92 to 2008–09) despite an aging population. Our findings support three recent European studies where both prevalence and incidence of dementia were reported to have decreased despite population aging [15–17]. Our findings are further corroborating a recent systematic review that found a declining incidence of dementia in high income countries [36]. Our findings are somewhat different than recent Canadian studies using administrative data that found an increasing prevalence of dementia [8–10]. However, two of those studies also reported a decreasing incidence of dementia. We found that men had a higher prevalence of cognitive impairment in the earlier CSHA study whilst women reported a higher prevalence of cognitive impairment in later CCHS–HA study. Our findings are consistent with recent studies in Spain and Japan in which men and women reported different prevalence of cognitive impairment [37, 38]. Our finding is however at odds with what Mathews et al. [15] who in a comparative study of dementia prevalence found that women were consistently more likely to report higher dementia prevalence compared to men. Freedman

**Table 6. Summary of risk and protective factors for cognitive impairment in the CSHA (1991–92) and the CCHS-HA (2008–09) study cohorts and stratified by gender.**

| Risk/Protective factor | Total population | | By gender | | | |
| --- | --- | --- | --- | --- | --- | --- |
| | CSHA | CCHS-HA | CSHA | | CCHS-HA | |
| | | | Male | Female | Male | Female |
| Increasing age | – | – | – | – | – | – |
| Being Female | + | ++ | NA | NA | NA | NA |
| Higher Education | ++ | ++ | ++ | ++ | + | ++ |
| "White"- ethnic/racial identity | ++ | ++ | ++ | ++ | ++ | ++ |
| Urban residence | + | + | NA | NA | NA | NA |
| Heart Disease | + | + | NA | ++ | + | + |
| Being single | – | NA | – | – | -NS | NA |
| Diabetes | NA | -NS | NA | NA | NA | -NS |
| Vision problems | – | NA | – | – | + | NA |
| Arthritis | ++ | NA | ++ | ++ | NA | -NS |
| High blood pressure | ++ | + | NA | ++ | -NS | ++ |
| Hearing problems | – | – | – | – | – | – |
| Parkinson | – | -NS | – | NA | – | NA |
| Stroke | – | – | – | – | – | -NS |
| SRH (poor) | – | – | – | – | – | – |

*[Significant risk factor (–), risk factor but not significant (-NS), protective factor with weaker strength of relationship (+), protective factor with stronger strength of relationship (++), not applicable (NA), Self-rated health (SRH)].

et al. using national survey data reports for the USA notes short term decline in the prevalence of probable dementia over the years 2011–15 [39]. The decline is attributed to the changing age and educational composition of the national population.

Also, our study found that even though there was a general decrease in cognitive impairment, the effect of the reduction was more prominent in men than in women. Both men and women in all age groups reported a decrease in cognitive impairment though the effect of the decrease was more significant for men than women. Our finding is at odds with an earlier finding of little or no differences in the prevalence of cognitive impairment between men and women [40].

We found an association between several modifiable and non-modifiable risk factors and cognitive impairment in our multivariable analyses. Firstly, we found five common risk factors for both study cohorts. Those who were older, had poorly rated their health, had suffered a stroke, had Parkinson's disease, and had hearing problems were more likely to report cognitive impairment. The above finding is consistent with earlier reports in the literature [22, 37, 41–43].

Second, we found five common protective factors of cognitive impairment in both cohorts. These include cultural or racial background ("white"), area of residence, high blood pressure, heart disease, and higher educational attainment. These findings are consistent with recent literature [43]. Meng, X. and D'Arcy C, [44] in a large systematic review found that low education increases the risk of dementia. We found that "whites" were less likely to report cognitive impairment compared to other racial/ethnic groups. In the United States, blacks were more likely to report cognitive decline compared to whites [40]. A systematic review using 14 longitudinal population-based studies of cognitive aging in 12 countries on 5 continents similarly found that Asians had a faster decline in cognition compared to whites [45]. The finding in this study that urban residents were less likely to report cognitive impairment is consistent with previous reports [9]. We found that being female was a protective factor against cognitive impairment. This finding is consistent with an earlier study where women were found to have performed better than men in both verbal and memory tests [45]. A possible explanation for the lack of sex differences between men and women is that both are afforded equal educational opportunities in Canada. Indeed, women usually outperform men scholastically. Our somewhat surprising protective factors of high blood pressure and heart disease may reflect the nature of the question asked (have you been told by a doctor that you have high blood pressure and/or heart disease?), and the fact that Canada has a universal publicly funded health care system and that these conditions are being effectively managed.

Furthermore, we found some contrasting findings between study cohorts. Four factors of marital status, diabetes, arthritis, and vision health problems produced divergent results for each cohort. There were risk factors and protective factors that were unique to each of the cohorts. For instance, we found that marital status (single or never married) was significantly associated with cognitive impairment in the CSHA cohort but was not a risk factor in the CCHS–HA cohort. The CSHA finding is consistent with earlier studies [43, 46]. Lipnicki et al. [46] reported that married compared to single status was a protective factor for the decline in executive function and reduces the risk of cognitive impairment and the vice versa. However, the changes in Canada that have occurred for marriage and family composition may be a possible explanation of the change in protection afforded by being married. Somewhat similarly, respondents with diabetes in the CCHS–HA cohort were more likely to report cognitive impairment, but this was not so in the CSHA cohort. This CCHS–HA finding is in keeping with what has been reported in other studies [37, 38, 43]. An explanation could be the recent increase in the incidence and prevalence of Type 2 diabetes in Canada, one of the fastest-growing diseases in the country. In the CSHA cohort arthritis was a protective factor for cognitive

impairment. This is consistent with an earlier study where those with arthritis had 24% higher odds of developing cognitive impairment [41]. Although there is some thought that NSAIDS such as ibuprofen typically used to treat arthritis may play a protective role in the development of dementia, the reason(s) for the change in the role of 'arthritis' is unclear.

In our sex-stratified analysis, we found that age, marital status, self-rated health, stroke, hearing problems, and vision problems were common risk factors between males and females in the CSHA cohort. This has been previously reported [37, 41–43]. We also found race/ethnicity, education, and arthritis as shared protective factors for cognitive impairment for both males and females. This finding is consistent with several other studies [16, 37, 40, 42, 44, 45]. Protective factors of high blood pressure and heart disease were uniquely negatively associated with cognitive impairment for females only. Also, Parkinson's disease was uniquely positively associated with cognitive impairment for males only. In contrast, Lipnicki et al. [46] report more physical activity and smoking as unique risk factors for cognitive impairment in men only.

Surprisingly, diabetes which is generally thought to be a risk factor of cognitive impairment was not associated with cognitive impairment in the CSHA cohort. Other studies have reported a significant association between diabetes and cognitive impairment [37, 41, 43]. In the CSHA sample, there were no risk factors for cognitive impairment that were either specific to males or females.

In the CCHS–HA, we found age, self-rated health, stroke, and hearing problems as shared risk factors of cognitive impairment cohort for both sexes. This is in line with what has been previously reported in the literature [37, 41, 43]. In this current study heart disease, education and race were shared protective factors. Other studies have previously reported similar findings [16, 37, 40, 42, 44, 45].

We also found marital status and Parkinson's disease as unique risk factors for males only in the CSHA sample. Our finding is consistent with what Yen et al. [43] reported where being single was associated with higher odds of developing cognitive impairment. Other risk factors such as diabetes and arthritis were found to be uniquely associated with cognitive impairment for females only in the CCHS–HA sample. This is at odds with Lipnicki et al. [46] who report that men rather than women were at a reduced risk of cognitive impairment or dementia if they had diabetes. Additionally, that same study reported a significant association between cognitive impairment and arthritis in males but not females which contrasts with our finding.

Our current findings give suggestive evidence that sex differences exist in the association between some predictor variables and cognitive impairment. In the CSHA sample, the effect of common risk factors of cognitive impairment in both sexes is more prominent in males than females. We reported that although both males and females reported shared risk factors of cognitive impairment, males had higher odds of cognitive impairment different from females on most of the risk factors investigated. Similarly, the effect of the predictor variables on the outcome in the CCHS–HA sample is more prominent in females compared to males. Females in this cohort had higher odds of reporting cognitive impairment than males in most of the risk factors measured.

We also found that modifiable risk factors of cognitive impairment changed over time by each sex. Whereas there were no unique risk factors of cognitive impairment in the CSHA cohort for females only, over time, protective factors in the earlier CHSA survey such as diabetes and arthritis became risk factors for female only. Also, our study found that unique risk factors of cognitive impairment for males only increased from one to two. Parkinson's disease was the unique risk factor for males only in the CSHA cohort, but marital status became an additional risk factor for males only in the CCHS–HA cohort.

## Strengths and limitations

The major strength of this study is the use of nationally representative and large national population-based samples of the Canadian senior population to estimate the prevalence of cognitive impairment at two points in time almost two decades apart. To the author's knowledge, this study is the first of its kind to use population-based study samples to examine the prevalence of cognitive impairment on a national scale in Canada. Most studies conducted in the country in the past were either province-specific or point prevalence estimates. Our study provided that comparative aspect which is lacking in the literature.

Another strength of our study is its ability to establish the age-cohort effect relationship between cognitive impairment and risk and protective factors as well as its prevalence two distinct points in time. Our study has explicitly established that more recent generations or birth cohorts were less likely to report cognitive impairment compared to earlier generations.

We also examined sex differences in both the prevalence and risk factors of cognitive impairment. This allows for sex-specific interventions to be tailored towards specific groups where it is most needed. Also, the CHSA and CCHS–HA study cohorts were among the few population-based studies in Canada to have specifically used survey instruments to measure cognition and to shed light on cognitive impairment or dementia in Canadian adults.

Despite these strengths, the study has some limitations. First, in our CSHA sample, the issue of imperfect sensitivity arises. Our analyses used the community sample to estimate both prevalence and predictors of cognitive impairment. However, the study recorded a high sensitivity value of 98.6% at baseline (CSHA-1) in the 3MS screening process, we cannot be sure that those deemed cognitively normal were not added to mild cognitive impairment cases.

Second, non-clinical measures of cognitive functioning were used in the CCHS−HA Cognition Module unlike the 3MS used in the CSHA sample. This is problematic because a clinical assessment is necessary to measure the sensitivity and specificity of a screening test in the cognitive decline or dementia identification process which is not available in the Cognition Module.

Third, our study could not include other important risk factors of cognitive impairment such as traumatic brain injury, obesity, smoking status, depression, sleep disturbances, hyperlipidemia and known protective factors such as physical activity, income, Mediterranean diet, cognitive training, moderate alcohol consumption, and social engagement. This is because some of these factors were not part of the CSHA sample which is the baseline data used for our comparison. We used variables for which data were available in both study samples.

## Conclusion

Our study provides some possible evidence of a reduction in the prevalence of cognitive impairment among community-dwelling Canadians in the context of an aging population. It reinforces the suggestion that although the increased prevalence of cognitive impairment could have been influenced by many factors such as survival after stroke, vascular incidents and diabetes, the decrease prevalence recorded in our study may be due to improvement in the prevention and treatment of vascular, stroke and hypertension morbidity and the increasing higher levels of education currently in the Canadian population [47]. Our results also provide evidence regarding how different experiences experienced by successive cohorts produce different patterns of disease risk in these generations and highlights the importance of cohort effects in public health prevention and treatment strategies. The study found that sex differences exist in the etiology of cognitive impairment and that these etiologies may change over time. We recommend a future longitudinal population-based study that looks at the associations found in this study. From a prevention point of view, future studies should focus on the

effectiveness of existing interventions to establish the extent to which they are meeting the demands of various segments of the population.

## Supporting information

**S1 File. Study questionnaire.**
(PDF)

## Author Contributions

**Conceptualization:** Batholomew Chireh, Carl D'Arcy.

**Data curation:** Batholomew Chireh.

**Formal analysis:** Batholomew Chireh.

**Investigation:** Batholomew Chireh, Carl D'Arcy.

**Methodology:** Batholomew Chireh.

**Supervision:** Carl D'Arcy.

**Validation:** Batholomew Chireh, Carl D'Arcy.

**Writing – original draft:** Batholomew Chireh.

**Writing – review & editing:** Carl D'Arcy.

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
