## [Decision Letter · Decision Letter 0]

23 Jun 2020

PONE-D-20-05879

A comparison of the prevalence of and modifiable risk factors for cognitive impairment among Canadian seniors over two decades, 1991–2009.

PLOS ONE

Dear Chireh,

Thank you for submitting your manuscript to PLOS ONE. After careful consideration, we feel that it has merit but does not fully meet PLOS ONE’s publication criteria as it currently stands. Therefore, we invite you to submit a revised version of the manuscript that addresses all the points raised during the review process.

We look forward to receiving your revised manuscript.

Kind regards,

Gianluigi Forloni

Academic Editor

PLOS ONE

Journal Requirements:

Additional Editor Comments (if provided):

Reviewers' comments:

Reviewer's Responses to Questions

**Comments to the Author**

1. Is the manuscript technically sound, and do the data support the conclusions?

Reviewer #1: No

Reviewer #2: Yes

2. Has the statistical analysis been performed appropriately and rigorously? 

Reviewer #1: No

Reviewer #2: Yes

3. Have the authors made all data underlying the findings in their manuscript fully available?

Reviewer #1: No

Reviewer #2: Yes

4. Is the manuscript presented in an intelligible fashion and written in standard English?

Reviewer #1: Yes

Reviewer #2: Yes

5. Review Comments to the Author

Reviewer #1: Thank you to the authors for their submission. My main critique is that the data do not support the conclusions. The main problem with comparing cognitive impairment between the CSHA and CCHS is that different tests are used in the different surveys. The CSHA administered the modified mini-mental state examination (3MS) and the CCHS utilized separate tests of immediate and delayed recall, animal naming and the mental alternation test. A derived variable was created classifying individuals as impaired if they scored in the “lowest cognitive functioning category” in any of the tests. No evidence is presented that this measure of cognitive impairment is equivalent to scoring less than or equal to 77 on the 3MS (the definition used in the CSHA). Thus direct comparisons cannot be made and the main conclusions suggesting the data support a birth cohort effect with decreasing prevalence over time should not be made.

It was reported that there was “a significant decrease in the prevalence of cognitive impairment in all age categories with the deceased being most pronounced among those 85+ with prevalence decreasing from 46 % and 37 % in men and women respectively in 1991-92 to 6.3% and 7.5 % respectively in 2008-09.” Such an enormous reduction does not seem plausible and is likely reflective of differences in testing and scoring rather than a true difference in prevalence. Implausibly, age did not seem to affect the prevalence of cognitive impairment in the CCHS (Table 2). Using men as an example, 9.2% of men age 65-74 years had cognitive impairment, compared to 8.4% of men aged 74-85 years, and only 6.3% of men aged 85+ years. In the CCHS, was cognitive impairment defined relative to people in the same age category, or across all age groups?

The authors could consider comparing results from the CCHS (2008-09) with those of the Canadian Longitudinal Study on Aging (CLSA) (2015 onwards) because both use the same cognitive measures. Furthermore, the data would be more relevant to contemporary readers since the CLSA is an ongoing study.

The comparison of differences in risk factors for cognitive impairment between the earlier CSHA and later CCHS is potentially valid since associations are being compared rather than prevalence. However, it appears that different statistical models were used for the different studies, because some predictors were not significant (defined as p<0.2) in univariable analyses and thus not carried over to multivariable models. I would suggest using the same models in which all predictors thought to be potential confounders are kept in both models. There are thousands of participants in both samples so there is no need to make the models more parsimonious to reduce the risk of type II error.

There is no information about missing data. The authors should present a table comparing patterns of missingness for each of the surveys (i.e., tables that compare the percentage of participants with missing and complete data, and the relevant characteristics of participants with missing and complete data). There needs to be a justification of the use of multiple imputation, i.e., a compelling argument that the data are missing completely at random or missing at random. A sensitivity analysis could have been performed comparing results using multiple imputation and complete case analysis. There was very little information about the methodology used for multiple imputation (e.g., multiple imputation model specifications).

It was not clear how survey weights were generated by the surveys analyzed in the paper, or how they were incorporated by the authors in their statistical methodology.

Some conclusions presented in the discussion seemed farfetched. For example, the authors wrote that “the high blood pressure and heart disease we found to be protective factors in the later CCHS-HA survey no doubts reflect the effective treatment and management of these chronic diseases in Canada over several decades,” without further explanation. I do not understand why treatment of high blood pressure or heart disease would lead to better cognitive outcomes. At best, perfectly treated hypertension or cardiovascular disease would eliminate the risk these conditions pose to the central nervous system. Cardiovascular disease, of course, is not perfectly treated, and it is still the second leading cause of death in Canada (https://www.canada.ca/en/public-health/services/publications/diseases-conditions/report-heart-disease-Canada-2018.html). Stroke was a risk factor for cognitive decline in the study, and stroke is a sequelae of cardiovascular disease. These inconsistencies needed a more thoughtful explanation.

Diabetes was found to be a risk factor for cognitive impairment in 2008-09 but not in 1991-92. It was argued that the discrepancy “could be explained by the recent increase in the incidence and prevalence of Type 2 diabetes in Canada,” but an increase in prevalence would not explain a change in the strength of association between the two variables.

There was minimal discussion of hearing loss in the introduction, discussion and conclusion even though it was found to be a consistent predictor of cognitive impairment and the 2017 Lancet Commissions paper on risk factors for dementia found that it had a population attributable fraction greater than any other potentially modifiable risk factor.

Throughout the manuscript, “multivariate” is used instead of the correct “multivariable.”

Reviewer #2: Overall, this is an important study, highlighting that in the context of aging, countries around the world should pay attention to cognitive impairment and the growing problem of dementia and Alzheimer. The method of this manuscript is more appropriate, and the conclusions reached are also very policy-relevant.

The manuscript also has certain deficiencies in the following aspects, which need to be revised or strengthened. For example,

1) Government or social organizations' data on the prevalence of cognitive impairment, dementia and Alzheimer's disease in Canadian elderly population should be clearly listed. It should be compared with the results of the cognitive impairment of the elderly measured in this paper.

2) The cross-sectional CSHA database and CCHS-HA database are used to estimate the incidence of cognitive impairment in the elderly in Canada, which has certain defects. It is necessary to introduce in detail the representativeness of these two data, especially in the representativeness of the national sample survey of the elderly in Canada. A representative introduction to these two data is essential. Related descriptions can be added to the manuscript.

3) Since the two databases use different tools/scales for measuring cognitive impairment of the elderly, is it scientifically feasible to compare the cognitive impairment of the elderly? Need to add relevant literature.

4) In terms of risk factors for cognitive impairment, some important factors are not included in the regression model, and it is recommended to add. Such as health behaviors, nursing services, rehabilitation training, etc. In this case, the corresponding policy intervention is more meaningful.

6. PLOS authors have the option to publish the peer review history of their article (what does this mean?). If published, this will include your full peer review and any attached files.

Reviewer #1: No

Reviewer #2: No

---

## [Author Response · Author response to Decision Letter 0]

24 Aug 2020

Response to review comments

Dear Professor Joerg Heber

Re: Response to Reviewer’s Comments

We appreciate the reviewer’s suggestions and comments on our manuscript and have carefully gone through the details and made sure the revised manuscript fully addresses the editor’s and reviewer’s comments.

Below we have outlined our responses in the attached Response to Review Comments.

The authors thank the reviewers for bringing these matters to our attention, we hope we have successfully addressed them.

Furthermore, because of those comments we have substantially revised the whole manuscript.

Response to Review Comments 

Specifically, the Reviewer made the following comments and we have made the following changes to the manuscript: (full details of the Reviewer’s comments are shown below):

Have the authors made all data underlying the findings in their manuscript fully available?

In response we have clearly indicated how this publicly available survey data can be accessed. We have written in the text of the manuscript as follows page 30:

Data Access.

Access to the CSHA data was provided by Canadian Study of Health and Aging - available at http://www.csha.ca.

Data access to the CCHS-HA Cognition Master File was provide by Statistics Canada through their Research Data Centres program that provides data access to Master File survey data to bona fide researchers upon application. This research was conducted at Saskatchewan Research Data Centre a part of the Canadian Research Data Centre Network (CRDCN). This service is provided through the support of University of Saskatchewan, the Canadian Foundation for Innovation, the Canadian Institutes of Health Research, the Social Science and Humanity Research Council, and Statistics Canada. 

Reviewer #1 

Reviewer Comment 1. 

The main problem with comparing cognitive impairment between the CSHA and CCHS is that different tests are used in the different surveys. The CSHA administered the modified mini-mental state examination (3MS) and the CCHS utilized separate tests of immediate and delayed recall, animal naming and the mental alternation test. A derived variable was created classifying individuals as impaired if they scored in the “lowest cognitive functioning category” in any of the tests. No evidence is presented that this measure of cognitive impairment is equivalent to scoring less than or equal to 77 on the 3MS (the definition used in the CSHA). Thus direct comparisons cannot be made and the main conclusions suggesting the data support a birth cohort effect with decreasing prevalence over time should not be made.

Response

We have in this manuscript acknowledged that different tests were used in the two respective cohort studies conducted some 16 to 18 years apart. There is no direct measure of comparing these two scales to the effect that a score of 77 on the 3MS is equivalent to the score of 1+ on the Cognition Module of the CCHS—HA 2009, nor could there be unless the same subjects were administered both tests at the same time or close in time. However, what both the 3MS and the Cognition Module claim is that they are screening tests for cognitive impairment. So, either these tests measure some degree of cognitive impairment or they don’t. The cut-off score for the 3MS is well established. The Cognition Module is also designed to assess cognitive impairment. Findlay et al. (2010) who developed and validated the 5 categories (0 to 4) for the Cognition Module used in our analysis conclude that “ These categories can be used in future work on cognitive functioning based on the CCHS-Heathy Aging.” (Abstract) The reviewer gives the impression that their main objection to the use of the Cognition Module to measure cognition impairment does not show an increase level of cognitive impairment. However, we would point out that we used the most minimal level of impairment in the Cognition Module which should have resulted in increased level of cognitive impairment but this did not result in a higher prevalence impairment in the study sample. While it would be tidier and easier if both studies had used the same cognitive test, we do not feel that there is anything wrong in comparing two tests that purport to measure the same thing - cognitive impairment - but recognize that it is not a perfect comparison, either the test measure cognitive impairment or they do not..

We would also note that the fact that there are data indicating a decline in dementia incidence in Canada. The Public Health Agency of Canada Infobase which looks at health care administrative data that includes data on Canadians in institutions shows a decline in dementia see below 

See https://health-infobase.canada.ca/ccdss/data-tool/?DDLV=16&DDLM=ASIR&sexB=B&yrB=2015&VIEW=0

It should be re-iterated that the above includes data on people in various kinds of institutions whereas the sample survey data we analyzed only looked at community-dwelling residents who we would expect to be more cognitively intact.

There is also other studies that report declines in dementia incidence in various countries, Rohr et al 2018. Kosteniuk et al 2016 using administrative data for institutional and community-dwelling residents report a decline in incidence in the Canadian Province of Saskatchewan. 

Reviewer Comment 2

It was reported that there was “a significant decrease in the prevalence of cognitive impairment in all age categories with the deceased being most pronounced among those 85+ with prevalence decreasing from 46 % and 37 % in men and women respectively in 1991-92 to 6.3% and 7.5 % respectively in 2008-09.” Such an enormous reduction does not seem plausible and is likely reflective of differences in testing and scoring rather than a true difference in prevalence.

Response

 We acknowledged this difference, but we are unable to assign any reason to the enormous reduction in cognitive impairment prevalence since that is exactly what the data revealed.

Reviewer Comment 3

Implausibly, age did not seem to affect the prevalence of cognitive impairment in the CCHS (Table 2). Using men as an example, 9.2% of men age 65-74 years had cognitive impairment, compared to 8.4% of men aged 74-85 years, and only 6.3% of men aged 85+ years. In the CCHS, was cognitive impairment defined relative to people in the same age category, or across all age groups?

Response

The authors acknowledged this finding. However, it should be noted that although age in general is a risk factors for cognitive impairment, it might not necessarily follow a linear trend. Similar findings were earlier reported by Mathews et al. 2013 in a two-decade comparison of prevalence of dementia in individuals aged 65 years and older (65-75 with dementia prevalence of 59.7% vs 75-84 with dementia prevalence of 19.6% among men) and in women reported (74-85 with dementia prevalence of 62.7% vs 85+ with dementia prevalence of 56.4%). 

Reviewer Comment 4

 The authors could consider comparing results from the CCHS (2008-09) with those of the Canadian Longitudinal Study on Aging (CLSA) (2015 onwards) because both use the same cognitive measures. Furthermore, the data would be more relevant to contemporary readers since the CLSA is ongoing.

Response

The reason for making the CHSA – CCHS-HA comparisons is that these studies are separated by 16 to 18 years between 1991-92 and 2008-09 a period of substantial change in Canada and period that saw a substantial aging of the Canadian population. As we note in the text of the manuscript: “…that the CSHA cohort is composed of individuals born prior to 1926 and who no doubt experienced the Great Depression and WWII whereas the CCHSA-HA 65+ cohort was born prior to 1943 had exposure to a substantially different social and economic environment.”

To compare the CCHS 2008-09 to the CLSA 2015+ would be comparing studies that occurred some 6 year apart we wonder how much change would have occurred in that time except the 2015 sample would see the beginning of the “baby boom” entering into the 65+ age group. Perhaps it would be more interesting to compare across all 3 studies. 

However, the point of our study is to assess whether the level of cognitive impairment had possibly decline in the Canadian population as has been report for other jurisdictions. Since the CHSA study has been used as the basis to project the future of the need for dementia care (the grey tsunami of care) in the Canadian population we think it is imperative to study changes in dementia incidence over time. We do not think the baseline CHSA should be ignored. Canadian data show substantial declines in morbidity and mortality for heart disease and stroke over the last several decades given the strong connection between cardio- and cerebro- vascular diseases it is not unthinkable that a decline may have occurred in the occurrence of dementia in the Canadian population.

Reviewer Comment 5

 The comparison of differences in risk factors for cognitive impairment between the earlier CSHA and later CCHS is potentially valid since associations are being compared rather than prevalence. However, it appears that different statistical models were used for the different studies, because some predictors were not significant (defined as p<0.2) in univariable analyses and thus not carried over to multivariable models. I would suggest using the same models in which all predictors thought to be potential confounders are kept in both models.

Response

We followed standard recommended practice excluding in the multivariate analysis factors which had unadjusted p<0.20 in the multivariate analysis. Also guided by previous literature, we added all variables in the multivariable analysis considered as plausible risk factors for cognitive impairment regardless of significance level from the univariable stage. Those that were insignificant after the multivariable analysis were left out afterwards. 

Reviewer Comment 6

 There is no information about missing data. The authors should present a table comparing patterns of missingness for each of the surveys (i.e., tables that compare the percentage of participants with missing and complete data, and the relevant characteristics of participants with missing and complete data).

Response

This information is presented in figure 1. It highlighted the cohort derivation process that clearly stipulated how many missing values were eliminated from the analysis. Besides, we conducted multiple imputation to account for the missing values in this study.

Reviewer Comment 7

 A sensitivity analysis could have been performed comparing results using multiple imputation and complete case analysis. There was very little information about the methodology used for multiple imputation (e.g., multiple imputation model specifications).

Response

 Because we conducted multiple imputation, we did not see the need to compare that with that of complete cases given that the missing values were retrieved through multiple imputation. This is clearly stated in page 10 of the manuscript as;

We performed multiple imputations to adjust for missing risk factor values and to prevent selection bias and loss of information. The imputations were generated using the chained equations procedure in STATA. All covariates and the outcome variable were included in the imputation model to ensure accurate estimation of missing values. After the imputation process, we retrieved all the missing values in both samples for the model building.

Reviewer Comment 8

 It was not clear how survey weights were generated by the surveys analyzed in the paper, or how they were incorporated by the authors in their statistical methodology.

Response

We could not utilise weighting in our analysis since the weighting variable was not provided in the CSHA data we accessed. Therefore, for the sake of uniformity, weighting was not conducted in both cohort samples.

Reviewer Comment 9

 Some conclusions presented in the discussion seemed farfetched. For example, the authors wrote that “the high blood pressure and heart disease we found to be protective factors in the later CCHS-HA survey no doubts reflect the effective treatment and management of these chronic diseases in Canada over several decades,” without further explanation. I do not understand why treatment of high blood pressure or heart disease would lead to better cognitive outcomes. At best, perfectly treated hypertension or cardiovascular disease would eliminate the risk these conditions pose to the central nervous system. Cardiovascular disease, of course, is not perfectly treated, and it is still the second leading cause of death in Canada (https://www.canada.ca/en/public-health/services/publications/diseases-conditions/report-heart-disease-Canada-2018.html). Stroke was a risk factor for cognitive decline in the study, and stroke is a sequelae of cardiovascular disease. These inconsistencies needed a more thoughtful explanation.

Response

On the contrary not farfetched treated HBP and heart disease means that the disease in under control. A treated BP of 130/70 is the same as an untreated BP of 130/70 is not? Does a treated BP of 130/70 not have the same health impact as an untreated BP of 130/70? 

Checking the cited web reference above shows that the age-standardized incidence of ischemic heart disease and acute myocardial infraction has declined substantially in the last 2 decades (Figure 2A), and if you look at the data over a longer time period the decline is more pronounced. Even the age-standardized incidence of heart failure has declined (Figure 2B).

The Canadian Chronic Disease Surveillance System also shows a substantial decline in age standardized ischemic heart disease.

We would note that stroke is not the sequalae (a condition which is the consequence of a previous disease or injury) of heart disease, they are separate diseases, but they may have several risk factors in common. But we do note that the ORs for stroke substantially decrease between CHSA and CCHS-HA.

Finally we note in the manuscript that:

“Our somewhat surprising protective factors of high blood pressure and heart disease may reflect the nature of the question asked (have you been told by a doctor that you have …?), and the fact that Canada has a universal publicly funded health care system and that these conditions are being effectively managed” (p18)

Reviewer Comment10

 Diabetes was found to be a risk factor for cognitive impairment in 2008-09 but not in 1991-92. It was argued that the discrepancy “could be explained by the recent increase in the incidence and prevalence of Type 2 diabetes in Canada,” but an increase in prevalence would not explain a change in the strength of association between the two variables. 

Response

Diabetes is a well-established risk factor for dementia, increasing prevalence could well increase the strength of the association.

Reviewer Comment 11

There was minimal discussion of hearing loss in the introduction, discussion and conclusion even though it was found to be a consistent predictor of cognitive impairment and the 2017 Lancet Commissions paper on risk factors for dementia found that it had a population attributable fraction greater than any other potentially modifiable risk factor.

Response

Thank you for bringing this point to our attention, we would also add the more recent Lacet Report on Dementia (2020) list hearing loss in midlife as a significant risk factor for dementia. The survey data examined here deal with populations that are 65+. But we do now note in the text of the manuscript the above point. 

Reviewer Comment 12

Throughout the manuscript, “multivariate” is used instead of the correct “multivariable.”

Response

Although we have checked its usage and it appears that either word is acceptable, we have changed multivariate to multivariable in the entire manuscript to reflect what is suggested.

Reviewer #2

Reviewer Comment 1

Government or social organizations' data on the prevalence of cognitive impairment, dementia and Alzheimer's disease in Canadian elderly population should be clearly listed. It should be compared with the results of the cognitive impairment of the elderly measured in this paper.

Response

In the discussion section of the paper we note: 

“Our findings are somewhat different than recent Canadian studies using administrative data that found an increasing prevalence of dementia [8-10]. However, two of those studies also reported a decreasing incidence for dementia.”(p 16) 

We further note that the administrative data for Canada on dementia cover both community-dwelling and institutional living population whereas our study sample only covered community-dwelling populations. 

Reviewer Comment 2

 The cross-sectional CSHA database and CCHS-HA database are used to estimate the incidence of cognitive impairment in the elderly in Canada, which has certain defects. It is necessary to introduce in detail the representativeness of these two data, especially in the representativeness of the national sample survey of the elderly in Canada. A representative introduction to these two data is essential. Related descriptions can be added to the manuscript.

 Response

We thank the reviewer this comment. But we should add that, both the CSHA and CCHA-HA are both national surveys conducted my Statistics Canada. Statistics Canada over the years has conducted highly representative surveys that are representative of the Canadian population and across the various regional blocks of the country. We have sufficiently highlighted the data collection process which reflects the representativeness of these surveys in pages 6 and 7 of this manuscript.

Reviewer Comment 3

 Since the two databases use different tools/scales for measuring cognitive impairment of the elderly, is it scientifically feasible to compare the cognitive impairment of the elderly? Need to add relevant literature.

 Response

We essentially dealt with this point under response 1 to Reviewer#1. We repeat it here 

We have in this manuscript acknowledged that different tests were used in the two respective cohort studies conducted some 16 to 18 years apart. There is no direct measure of the comparing these two scales to the effect that a score of 77 on the 3MS is equivalent to the score of 1+ on the Cognition Module of the CCHS—HA 2009, nor could there be unless the same subjects were administered both tests at the same time or close in time. However, what both the 3MS and the Cognition Module claim is that they are screening tests for cognitive impairment. So, either these tests measure some degree of cognitive impairment or they don’t. The cut-off score for the 3MS is well established. The Cognition Module is also designed to assess cognitive impairment. Findlay et al. (2010) who developed and validated the 5 categories (0 to 4) for the Cognition Module used in our analysis conclude that “ These categories can be used in future work on cognitive functioning based on the CCHS-Heathy Aging.” (Abstract) The reviewer gives the impression that their main objection to the use of the Cognition Module to measure cognition impairment does not show an increase level of cognitive impairment. However, We would point out that we used the most minimal level of impairment in the Cognition Module which should have resulted in increased level of cognitive impairment but this did not result in a higher prevalence impairment in the study sample. While it would be tidier and easier if both studies had used the same cognitive test, we do not feel that there is anything wrong in comparing two tests that purport to measure the same thing - cognitive impairment - but recognize that it is not a perfect comparison, either the test measure cognitive impairment or they do not..

We would also note that the fact that there are data indicating a decline in dementia incidence in Canada. The Public Health Agency of Canada Infobase which looks at health care administrative data that includes data on Canadians in institutions shows a decline in dementia see below 

See https://health-infobase.canada.ca/ccdss/data-tool/?DDLV=16&DDLM=ASIR&sexB=B&yrB=2015&VIEW=0

It should be re-iterated that the above includes data on people in various kinds of institutions whereas the sample survey data we analyzed only looked at community-dwelling residents who we would expect to be more cognitively intact.

There is also other studies that report declines in dementia incidence in various countries, Rohr et al 2018. Kosteniuk et al 2016 using administrative data for institutional and community-dwelling residents report a decline in incidence in the Canadian Province of Saskatchewan. 

Reviewer Comment 4

 In terms of risk factors for cognitive impairment, some important factors are not included in the regression model, and it is recommended to add. Such as health behaviors, nursing services, rehabilitation training, etc. In this case, the corresponding policy intervention is more meaningful.

 Response

Our study uses secondary analysis of already existing data, so we are limited by the variables collected in the original study,

We acknowledge this limitation in both the Measure and the Discussion sections of the manuscript.

In the Measures section we write”

“Both surveys collected information on the following predictors or covariates were therefore included in the analysis: sex (“male” and “female”), age (65–74, 75–84, 85+ years), area of residence (“rural” and “urban”), educational level (“Less than secondary”, “Secondary graduation”, “Some post-secondary graduation”, “Post-secondary graduation”), province (“Atlantic”, “Quebec”, “Ontario”, “Prairies”, “British Columbia”), marital status (“Married/common law”, “Widowed/divorced/separated”, “Single/never married”), cultural or racial background (“white” and “all other race/non-white”), self-rated health (“excellent/very good/good” and “fair/poor”), high blood pressure (“yes” and “no”), heart disease (“yes” and “no”), arthritis (“yes” and “no”), Parkinson disease (“yes” and “no”), diabetes (“yes” and “no”), stroke (“yes” and “no”), hearing problems (“yes” and “no”) and vision problems (“yes” and “no”).”

In the Discussion under Strengths and Weakness we write that:

“Third, our study could not include other important risk factors of cognitive impairment such as traumatic brain injury, obesity, smoking status, depression, sleep disturbances, hyperlipidemia and known protective factors such as physical activity, income, Mediterranean diet, cognitive training, moderate alcohol consumption and social engagement. This is because some of these factors were not part of the CSHA sample which is the baseline data used for our comparison. We used variables for which data was available for in both study samples.” (pp 21-22)

---

## [Decision Letter · Decision Letter 1]

12 Nov 2020

A comparison of the prevalence of and modifiable risk factors for cognitive impairment among community-dwelling Canadian seniors

PONE-D-20-05879R1

Dear Dr. Chireh,

We’re pleased to inform you that your manuscript has been judged scientifically suitable for publication and will be formally accepted for publication once it meets all outstanding technical requirements.

Kind regards,

Gianluigi Forloni

Academic Editor

PLOS ONE

Additional Editor Comments (optional):

Reviewers' comments:

Reviewer's Responses to Questions

**Comments to the Author**

1. If the authors have adequately addressed your comments raised in a previous round of review and you feel that this manuscript is now acceptable for publication, you may indicate that here to bypass the “Comments to the Author” section, enter your conflict of interest statement in the “Confidential to Editor” section, and submit your "Accept" recommendation.

Reviewer #2: All comments have been addressed

2. Is the manuscript technically sound, and do the data support the conclusions?

Reviewer #2: Yes

3. Has the statistical analysis been performed appropriately and rigorously? 

Reviewer #2: Yes

4. Have the authors made all data underlying the findings in their manuscript fully available?

Reviewer #2: Yes

5. Is the manuscript presented in an intelligible fashion and written in standard English?

Reviewer #2: Yes

6. Review Comments to the Author

Reviewer #2: The author was very serious in the revision process and answered all my doubts. At the same time, the materials provided are also richer and meet my requirements. This is a valuable study, because in the context of global aging, cognitive dysfunction has become the most important factor affecting human health and longevity. The study provides more consensus on dementia. A certain degree of innovation also provides reference value for the introduction of relevant intervention policies. Based on. I think that the author's modification meets PLOSONE's publication requirements, and I agree that PLOSONE accepts the manuscript. It is also suggested that PLOSONE can accept more related manuscripts on elderly population dementia, or set up more related topics.

7. PLOS authors have the option to publish the peer review history of their article (what does this mean?). If published, this will include your full peer review and any attached files.

Reviewer #2: No

---

## [Editor Report · Acceptance letter]

18 Nov 2020

PONE-D-20-05879R1 

A comparison of the prevalence of and modifiable risk factors for cognitive impairment among community-dwelling Canadian seniors 

Dear Dr. Chireh:

I'm pleased to inform you that your manuscript has been deemed suitable for publication in PLOS ONE. Congratulations! Your manuscript is now with our production department. 

Kind regards, 

on behalf of

Dr. Gianluigi Forloni 

Academic Editor

PLOS ONE